# Using the Colloidal Method to Prepare Au Catalysts for the Alkylation of Aniline by Benzyl Alcohol

**DOI:** 10.3390/ijms241914779

**Published:** 2023-09-30

**Authors:** Luka V. Hare, Firdaus Parveen, James Cookson, Peter R. Ellis, Klaus Hellgardt, King Kuok (Mimi) Hii

**Affiliations:** 1Department of Chemistry, Imperial College London, Molecular Sciences Research Hub, 82 Wood Lane, London W12 0BZ, UK; 2Johnson Matthey, Blount’s Court, Sonning Common, Reading RG4 9NH, UK; james.cookson@matthey.com (J.C.); peter.ellis@matthey.com (P.R.E.); 3Department of Chemical Engineering, Imperial College London, Exhibition Road, London SW7 2AZ, UK; k.hellgardt@imperial.ac.uk

**Keywords:** Au-supported catalyst, structure–activity relationship, H-borrowing catalysis, alkylation of amine

## Abstract

Using the colloidal method, attempts were made to deposit Au NPs on seven different material supports (TiO_2_, α and γ-Al_2_O_3_, HFeO_2_, CeO_2_, C, and SiO_2_). The deposition between 0.8 and 1 wt% of Au NPs can be generally achieved, apart for SiO_2_ (no deposition) and α-alumina (0.3 wt%). The resultant sizes of the Au NPs were dependent on the nature as well as the surface area of the support. The catalytic activity and selectivity of the supported Au catalysts were then compared in the alkylation of aniline by benzyl alcohol. Correlations were made between the nature of the support, the size of the Au NP, and the H-binding energy. A minimum H-binding energy of 1100 μV K^−1^ was found to be necessary for high selectivity for the secondary amine. Comparisons of the TEM images of the pre- and post-reaction catalysts also revealed the extent of Au NP agglomeration under the reaction conditions.

## 1. Introduction

Secondary amines are an industrially important feedstock used in the synthesis of fine chemicals, surfactants, dyes, agrochemicals, functionalized materials, and biologically active compounds. Secondary amines are mostly derived from primary amines, either by reaction with a reactive alkyl halide or reductive amination with a carbonyl compound (Figure 1, routes 1 and 2, respectively). These reactions utilize stoichiometric amounts of toxic alkyl halides or hazardous hydridic reductants, which also generate stoichiometric amounts of by-products that require complex workup procedures for product purification.

In comparison, catalytic alkylation of amines using a primary alcohol (Figure 1, route 3), also known as ‘H-auto transfer’ or ‘hydrogen-borrowing’ reactions, is considered to be a ‘greener’ method for the synthesis of secondary amines. This methodology does not require extraneous reagents, and only water is produced as the by-product, resulting in a more sustainable and scalable process [1,2]. The reaction broadly follows the following elementary steps (Figure 2): (i) dehydrogenation/oxidation of alcohol to the corresponding aldehyde or ketone; (ii) condensation of the carbonyl group with amine to form an imine intermediate, which is (iii) reduced by the metal-hydride formed in step (i) to the amine product.

To date, a plethora of homogeneous and heterogenous metal catalysts have been reported for the H-auto transfer reaction, and the subject has been covered by a number of comprehensive reviews [3,4,5]. Often, the imine, R^2^CH=NR^1^, and the tertiary amine (R^2^CH_2_)_2_NR^1^ were observed as side products, necessitating the addition of a base to suppress these impurities. However, the inclusion of a base in such reactions can be problematic, as it is known that these types of reactions can indeed proceed under basic conditions without metal catalysts [6,7].

In our earlier work, we demonstrated that the reaction can be performed with very high selectivity in the absence of additives or bases by using a commercially available Au/TiO_2_ catalyst (AUROlite^TM^) in a packed-bed reactor [8]. Subsequently, we initiated a study to examine how metal support interactions (MSIs) may affect the catalyst performance in these reactions. At a very basic level, MSIs reduce the mobility of metal nanoparticles on the catalyst support and their tendency to agglomerate, enhancing catalytic activity and stability through the maintenance of particle sizes during catalytic turnover [9]. However, MSIs may also be attributed to the presence of special sites at the perimeter of metal particles, where the chemical and electronic properties of both the metal and support atoms can influence the adsorbed species, thus bestowing added catalytic activity [10]. In an earlier study by Ishida et al. [11], Au catalysts supported on nine metal oxides were prepared by different methods (milling, co-precipitation, and deposition-precipitation) and utilized in the reaction of aniline with benzyl alcohol. It was found that while basic and neutral supports promote catalytic activity (measured by alcohol conversion), the selectivity to secondary amine is attributed to the adsorption of aniline by hydrogen bonding with Lewis acidic sites on the surface of the metal oxides. However, the study did not include any analysis of Au nanoparticles (size and distribution), which can result from the different methods of operation.

In this work, we aim to prepare a series of Au NPs supported on different supports using the colloidal method in an attempt to study how the nature of the support may influence the size of the Au nanoparticles and their catalytic activity in H-auto transfer reactions.

## 2. Results and Discussion

### 2.1. Catalyst Preparation

The sol immobilization method (colloidal synthesis) was chosen for the preparation of the Au catalysts. The procedure deposits pre-formed Au NPs onto a solid support. As the metal nanoparticles are formed independently of the support, it can offer more consistent results in terms of control of the particle sizes [12]. The colloidal method was deployed successfully by Hutchings and co-workers to produce 1 wt% Au/TiO_2_ catalysts using polyvinyl alcohol (PVA) as a stabilizer [13]. In this procedure, the resultant catalyst particles were heated in water at 90 °C for 2 h to remove the water-soluble stabilizer (Figure 1). As the catalyst was not exposed to very high temperatures, the size of the Au nanoparticles was preserved (2.9 nm). However, calcination of the catalyst at temperatures ≥ 300° can also lead to significant sintering of the metallic NP.

In the present work, we deployed the colloidal method to deposit Au NPs onto seven different supports: titania (TiO_2_, P25), carbon (C), ferric oxyhydroxide (HFeO_2_), ceria (CeO_2_), γ-alumina (γ-Al_2_O_3_), and silica (SiO_2_). Following the treatment with water, the ‘as-prepared’ catalysts were air-dried in an oven at 100 °C for 24 h. The samples were subjected to analyses by ICP-OES (wt% Au), TEM (average particle size and distribution are shown in Figure 2—left column), and BET (surface area, Appendix A). To assess the removal of the PVA stabilizer, the amounts of residual C on the catalysts were also quantified by combustion analysis. The results are summarised in Table 1.

With the exception of SiO_2_ (entry 7), the deposition of AuNP was achieved with varying degrees of success on the supports. The method successfully reproduced Au/TiO_2_ with very similar properties as that reported before, with close to 1 wt% loading of Au and average particle sizes of 2.9 nm (entry 1) with a narrow distribution (Figure 2). The method also provided an even distribution of very small Au NPs on γ-Al_2_O_3_ (2.6 nm), with a lower 0.6 wt% catalyst loading (entry 2). In contrast, the deposition of Au NPs on α-Al_2_O_3_ was poor, with very low loading (0.3 wt%) and a very broad distribution of large particle sizes (entry 3 and Figure 2). We attribute this to the small surface area afforded by this material, with limited availability of sites for the pre-formed Au NPs to adhere to. The supported Au NPs on the surface are also likely to be in close proximity, which can agglomerate, even under mild thermal treatment conditions, to form large NPs with very wide size distributions (Figure 2). The removal of PVA stabilizer from these three catalysts using the hydrothermal treatment was found to be successful, with <0.1 wt% of carbon remaining.

The surface of active carbon comprises polycyclic aromatic groups to which metallic nanoparticles can anchor. However, it was known that the preparation of Au/C by deposition–precipitation methods can be difficult due to the hydrophobic nature of carbon and the low density of surface OH groups [9]. Using the colloidal method, 0.9 wt% Au loading can be achieved (entry 5). The average particle size (3.4 nm) of the Au NPs is slightly bigger than that deposited on TiO_2_ and γ-Al_2_O_3_, but the particle size distribution is reasonably narrow (Figure 2). This shows that fairly good size control can be achieved by the method onto relatively unfunctionalized supports as long as there is a large surface area for the NP to deposit onto. In comparison, the availability of surface hydroxyl groups in HFeO_2_ allowed the target 1 wt% of Au to be achieved (entry 5). Although the average particle size of 3.7 nm was obtained, a wide particle size distribution between 2 and 10 nm was found (Figure 2), suggesting that substantial agglomeration has occurred during the drying process, likely to also be due to the poor distribution of NP on the small surface area. Last but not least, the preparation of Au/CeO_2_ using the colloidal method was previously described to produce Au NPs of around 3 nm [14]. In this work, we were able to obtain 0.8 wt% of Au on CeO_2_, but we were not able to establish the particle sizes due to the poor contrast between Au and the dense support on the TEM grid. For the two catalysts supported on HFeO_2_ and CeO_2_, 0.4 and 0.5 wt% of residual carbon can be found, signifying that the hot water treatment was not effective for the removal of PVA from these catalysts. Subsequently, these catalysts were calcinated at 200 °C under 5% H_2_-N_2_ to remove the residual stabilizer. The other Au catalyst supported on a reducible metal oxide—Au/TiO_2_—was also subjected to the same thermal treatment to provide a comparison.

### 2.2. Catalyst Activity

The catalytic activities of the Au catalysts for the H-auto transfer reactions were subsequently assessed. Using benzyl alcohol (**1**) and aniline (**2**) as model substrates, the evaluation was conducted in parallel under the same conditions (Figure 3). Apart from the expected product **3**, the reaction mixture may also contain reaction intermediates imine **4** and benzaldehyde **5**. Competitive formation of toluene (**6**) as a side-product is also possible, but this can escape detection as it is often employed as a reaction solvent; in this case, the quantification of **6** can be achieved by using 2-methyl-2-butanol as a solvent.

The catalytic results are presented in Table 2. Turnover frequencies (TOFs) were used to compare the catalyst activities to account for the different amount of Au deposited on each support. The performance of Au/TiO_2_ prepared by the colloidal method mirrors earlier results obtained with commercially available Au/TiO_2_ [8], with very good selectivity for the expected product **3** (entry 1). Au/γ-Al_2_O_3_ also performed very well, with very similar outcomes (entry 2). In contrast, the corresponding catalyst supported on the α-allotrope was practically inactive (entry 3), which is perhaps unsurprising, given the much larger particle sizes.

On the other hand, while Au supported on activated carbon and HFeO_2_ both afforded similar average NPs of 3.4 and 3.7 nm (Table 1, entries 4 and 5), the former is inactive compared to the moderate turnover obtained with Au/HFeO_2_, even in the presence of residual PVA (Table 2, entries 4 and 5). It is also interesting to see that the formation of the imine intermediate **4** was observed as the major product for the catalyst supported on activated carbon compared to the other metal oxides, which also suggests that the availability of surface hydroxyl groups is important for the H-transfer necessary to convert **4** to **3**. To test this further, H_2_ temperature-programmed desorption (H-TPD) studies were performed with the different Au catalysts. The approximate binding energies can be calculated by multiplying the peak integration values from H-TPD analysis by the peak desorption temperatures (μmol H_2_ per gram cat × peak integration). The values were subsequently plotted against the observed amine selectivity (Table 3 and Figure 3). Broadly speaking, there appears to be a direct correlation between the catalyst’s ability to bind to H_2_ and their catalytic activity, in the increasing order: Au/C, Au/α-Al_2_O_3_ (TOF < 15 h^−1^) < Au/HFeO_2_, Au/CeO_2_ (ca. 90 h^−1^) < Au/TiO_2_, Au/γ-Al_2_O_3_ (TOF between 170 and 180 h^−1^). This result reveals that a binding energy above 10 mV K^−1^ is required to maintain a high selectivity (>95%) for the secondary amine **3**.

Thermal treatment of Au catalysts supported on reducible metal oxides (including TiO_2_, HFeO_2_, and CeO_2_) a reducing atmosphere is known to induce strong metal–support interactions, which can affect the catalyst activity, either in a positive or negative way [10]. Conversely, thermal treatment of Au catalysts prepared by the colloidal method is also found to be very susceptible to agglomeration under thermal conditions, leading to reduced catalyst activity in CO oxidation reactions [13]. In the present study, calcination resulted in three different observations: The calcination of Au/TiO_2_ led to a decrease in catalyst activity but did not affect the selectivity for amine **3** (Table 2, entries 1 vs. 7), while the calcination of Au/FeO_2_ increased the catalytic activity but did not affect the selectivity (Table 2, entries 5 vs. 8). Last but not least, the calcination of CeO_2_ led to the deterioration of both reactivity and selectivity (entries 6 and 9). These results show that calcinating these catalysts (with corresponding changes in particle size) does not lead to any improvements in the selectivity of the process.

Indeed, agglomeration of the Au NPs may also occur during the catalytic turnover conditions. Following reactions at 180 °C for 30 min, the catalysts were recovered and subject to TEM analyses (Table 4 and Figure 2, right column). In all cases, Au particle sizes increased by between 16 and 29% for the metal oxide supports (Entries 1, 2, 3 and 5). In contrast, the recovered Au/C catalyst was found to be 1.5 times bigger than the ‘as-prepared’ catalyst (entry 4), with very dramatic changes in the particle distribution (Figure 2, right column). From this, we can surmise that one of the main roles of the (metal oxide) support is to maintain the particle size of the Au NP during the catalytic reaction.

## 3. Materials and Methods

### 3.1. General

Unless otherwise stated, all chemical precursors, solvents, and standards employed in this work were procured commercially and used without further purification. TiO_2_ support (P25-Degussa) was employed in this work, containing 25% anatase and 75% rutile phases, and all other catalyst supports were provided by Johnson Matthey plc. The conversion of substrates to products was monitored using an HP6890 Gas Chromatograph, equipped with a H_2_ flame ionisation detector and an HP5 Agilent column (30 m × 320 μm × 0.25 μm). The percentage of conversion and selectivity was determined by comparison with known standards, using calibration plots and 4-tert-butylphenol (50 mM in methanol) as an external standard. TEM images were captured at the Harvey Flowers Electron Microscopy Suite at Imperial College London, using a JEOL 2010 TEM instrument operated at 200 kV, with a probe current of 108 µA, and a Gatan Orios camera. XRD analysis was conducted at Johnson Matthey plc using a Bruker AXS D8 diffractometer. ICP-OES analysis was conducted at Johnson Matthey plc using a Perkin Elmer Optima instrument. Chemisorption studies (H-TPD) were performed at Johnson Matthey plc using Altamira AMI-200 apparatus.

### 3.2. Preparation of Catalysts

Preparation of Au/support [13]: A colloidal solution of Au was prepared by addition of poly(vinyl)alcohol (PVA, Mw 9000–10,000, 80% hydrolysed, 0.1 wt%) to HAuCl_4_·3H_2_O (0.50 mmol of Au) solution in 500 mL of H_2_O. A freshly prepared solution of aq. NaBH_4_ (13 mM) was then added to the mixture to form a dark-brown sol. After 30 min, the colloidal solution was added to the requisite support (e.g., TiO_2_) with vigorous stirring (500 rpm). The resulting material was collected by filtration and exhaustively washed with deionised H_2_O. The catalyst was dried in an oven at 100 °C for 24 h before it was transferred into a round bottom flask and heated in deionised H_2_O at 90 °C with vigorous stirring (500 rpm) for 2 h to remove the PVA [13]. The resulting solid was collected, washed thoroughly with deionised H_2_O, and dried in an oven at 100 °C for 24 h.

Thermal treatments (calcination): The ‘as-prepared’ catalysts (above) were reduced under a flow of N_2_/H_2_ (200 mL/min), using a Carbolite STF tube furnace, respectively. A temperature ramp of 10 °C/min^−1^ up to 200 °C was applied and held for 2 h before cooling to room temperature.

### 3.3. Catalyst Screening

An Endeavor^®^ catalyst screening system (Biotage) was employed in this part of the work. The reactor consists of 8 parallel reaction vessels with glass inserts (working volume 5 mL). Each reaction vessel was charged with a catalyst (0.9 mol% Au, average particle size 190 μm), and 2 mL stock solutions of aniline (0.5 M) and benzyl alcohol (0.5 M) in 2-methyl-2-butanol. The reaction vessels were sealed and purged with 3 cycles of N_2_ before being pressurised to 15 bar. The biphasic mixtures were stirred using paddles (250 rpm) and heated to 180 °C. After 30 min, the reaction mixtures were cooled to room temperature. The reaction aliquots were extracted, diluted, and analysed using a HP 6890 Gas Chromatograph equipped with an FID detector and an Agilent HP5 column (30 m × 320 μm × 0.25 μm). 1 μL of analyte solution was injected into the inlet, which was heated to 250 °C with a split ratio of 5:1. The system was operated under a constant pressure of 20 psi with an initial column temperature of 50 °C, held for 0.5 min, heated to 65 °C @2.5 °C/min, and finally to 200 °C @25 °C/min. The conversion of benzyl alcohol and selectivity of imines, amines, and other intermediates were calculated using known standards and 4-*tert*-butylphenol (50 mM in methanol) as an external standard. The benzyl alcohol conversion and product selectivity were calculated as follows:$$\begin{array}{cc} \%\;\mathrm{conversion}\;\mathrm{of} & \mathrm{benzyl}\;\mathrm{alcohol}\\ & =\frac{\mathrm{Initial}\;\mathrm{moles}\;\mathrm{of}\;\mathrm{BA}\;-\;\mathrm{moles}\;\mathrm{of}\;\mathrm{BA}\;\mathrm{in}\;\mathrm{reaction}\;\mathrm{mixture}}{\mathrm{Initial}\;\mathrm{moles}\;\mathrm{of}\;\mathrm{BA}\;}\times 100 \end{array}$$
$$\%\;\mathrm{Selectivity}=\frac{\mathrm{moles}\;\mathrm{of}\;\mathrm{products}\;\mathrm{in}\;\mathrm{reaction}\;\mathrm{mixture}}{\mathrm{moles}\;\mathrm{of}\;\mathrm{benzyl}\;\mathrm{alcohol}\;\mathrm{converted}\;}\times 100$$

## 4. Conclusions

In this work, attempts were made to deposit pre-formed Au NPs onto six different supports using the colloidal method, followed by a mild hydrothermal treatment to remove the PVA stabilizer and drying (temperatures of < 100 °C). The resultant material was found to contain varying amounts of Au (from 0–1 wt%), and the size and distribution of Au NPs are largely dependent on the available surface area, which has a pronounced effect on the agglomeration of deposited NPs, even under mild conditions. In this regard, γ-Al_2_O_3_ and TiO_2_ offered the smallest average particle sizes (2.6 and 2.9 nm) with a narrow distribution. Subsequently, the catalytic activity was found to be predominantly dominated by particle size—calcination of the reducible metal oxides did not lead to any significant enhancement in catalytic turnover. Agglomeration of Au during the catalytic turnover is highly dependent on the nature of the support, and this will also account for the different catalyst activities. Last but not least, a direct correlation between H_2_ binding efficiency and catalyst activity and selectivity can be observed. Overall, this preliminary study has helped us identify some important selection criteria for future catalyst design and development. These include a better method of catalyst preparation that can improve the thermal stability of the supported nanoparticles to minimize agglomeration under reaction conditions and the selection of materials with strong H-absorptivity, as established by H-TPD studies, to improve the catalyst activity and selectivity.

## Data Availability

All the research data generated from this study are contained within the manuscript.

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
