# Peer review of "Using the Colloidal Method to Prepare Au Catalysts for the Alkylation of Aniline by Benzyl Alcohol"

_ijms, 2023, doi:10.3390/ijms241914779_

Round 1
Reviewer 1 Report
This is an interesting work that can deserve publication in the International Journal of Molecular Sciences. However, I have the following comments that the authors should carefully implement in the revised manuscript prior to publication.
1) The abstract is too short and should be extended by including a description of the main findings of the study.
2) Introduction - The connection between the aim of the work and the literature gaps should be more deeply discussed, thus giving more strength to the reason for this work.
3) A colloidal suspension consisting of nano-sized ceria particles dispersed in a solution of acetic acid was also used to deposit metal-doped ceria on a structured SiC support (Topics in Catalysis, 2021, 64(3-4), pp. 256–269) and this should be acknowledged in the revised manuscript.
4) Results and discussion and Conclusions - The practical impact of the results obtained in this work should be better highlighted.
5) Conclusions - The authors should also give a structured outlook on future research work.
I’m willing to review the revised manuscript.
Author Response
(Reviewer's comments in bold)
This is an interesting work that can deserve publication in the International Journal of Molecular Sciences. However, I have the following comments that the authors should carefully implement in the revised manuscript prior to publication.
Response: We are grateful to the reviewer for the positive support and the suggestions to improve our manuscript.
1) The abstract is too short and should be extended by including a description of the main findings of the study.
Response: The abstract is expanded to include the main findings of the study.
2) Introduction - The connection between the aim of the work and the literature gaps should be more deeply discussed, thus giving more strength to the reason for this work.
Response: We thank the reviewer for this comment. We have supplemented the introduction accordingly.
3) A colloidal suspension consisting of nano-sized ceria particles dispersed in a solution of acetic acid was also used to deposit metal-doped ceria on a structured SiC support (Topics in Catalysis, 2021, 64(3-4), pp. 256–269) and this should be acknowledged in the revised manuscript.
Response: The reference provided by the reviewer led us to this article: “Synergy Between Ceria and Metals (Ag or Cu) in Catalytic Diesel Particulate Filters: Effect of the Metal Content and of the Preparation Method on the Regeneration Performance”? Unfortunately, we failed to see any relevance either in the preparation of the catalyst (does not include Au) or the application (autocatalysis)?
4) Results and discussion and Conclusions - The practical impact of the results obtained in this work should be better highlighted.
Response: We thank the reviewer for this comment. We have supplemented the conclusion section with the main findings of this work, and hope that this is sufficient.
5) Conclusions - The authors should also give a structured outlook on future research work.
Response: We thank the reviewer for these suggestions. These sections have been enhanced in the revised manuscript
Reviewer 2 Report
In my oppinion before possible publication in this Journal, this manuscript should be thoroughly revised concerning the subjects listed below
In the work, it was studied the catalytic activity of a series of materials consisting of Au Np´s supported onto five different metal oxides and on AC, in the alkylation of aniline by benzyl alcohol. The sizes of the adsorbed Au NP´s were preformed by using polyvinylalcohol as stabilizer.
Major points to revise;
1) According to the above, differences of the mean sizes of the adsorbed particles among the metal oxides used as supports (except in the case of Au--alphaAl2O3, suitably explained)are not significant. Thus, it could be think that the pore sizes of the metal oxides could be considered as a key parameter determining the sizes of the Au NP´s adsorbed. The last is also supported by in spite of large different surface areas of Au-Al2O3 (Table 1, entry 2) and Au-HFeO2 both exhibit similar adsorption capacity to Au. According to that it should be interesting to discuss on some possible relationship of de size of Au NP´s with pore distribution data of the different supports.
2) In the case of Au/C it should report the pore size distribution data of the support C. This could be usefull for a best understanding of the relative lower catalytic activity than the metal oxyde supports. This is due to (probably) a significant amount of the adsorbed Au NP´s should be placed into very narrow pores which either difficults the acces of the substrated to them or where these suffer from severe sterical restriction
3) Concerning the data of Table 4, (on the basis of hypothetical nature of the interaction of Au with the different supports) some tentative explanation should be done about larger agglomeration of Au/C NP´s relative to metal oxyde supports.
4) Neither in section 2.2 nor in Experimental section it is clarified if 30 min. in the kinetic studies was the equilibrium times. Differences in % conversion of Table 2 should reflect, in each of the cases, that different factors affect the mechanisms of the studied reaction determining different equilibrium times; e.g. in case of Au/C, expected adsorptivity of the aromatic substrates on activated carbon marks an important difference with metal oxide supports. This point deserves a short discussion the author should include in the manuscript.
Minor questions:
The following text of rows 20-21, "Secondary amines are mostly derived from primary amines, either by the reaction with a reactive alkyl halide or reductive amination with a carbonyl compound(Scheme 1, routes 1 and 2)" does not fully match with Scheme 1. Please revise.
In the text at foot of Table 1, the superscript 1 should be changed by a one, isn´t it?
On discussing the adsorptivity of Au (soft Lëwis acid) on AC, arene centres existing in the surface of the last, perhaps should be considered to act as active adsorption sites.
TEM images used for the counting of the sizes distribution of Au NP´s should be provided as supplementary information
Author Response
Reviewer comments in bold.
In my oppinion before possible publication in this Journal, this manuscript should be thoroughly revised concerning the subjects listed below
In the work, it was studied the catalytic activity of a series of materials consisting of Au Np´s supported onto five different metal oxides and on AC, in the alkylation of aniline by benzyl alcohol. The sizes of the adsorbed Au NP´s were preformed by using polyvinylalcohol as stabilizer.
Major points to revise;
1) According to the above, differences of the mean sizes of the adsorbed particles among the metal oxides used as supports (except in the case of Au--alphaAl2O3, suitably explained)are not significant. Thus, it could be think that the pore sizes of the metal oxides could be considered as a key parameter determining the sizes of the Au NP´s adsorbed. The last is also supported by in spite of large different surface areas of Au-Al2O3 (Table 1, entry 2) and Au-HFeO2 both exhibit similar adsorption capacity to Au. According to that it should be interesting to discuss on some possible relationship of de size of Au NP´s with pore distribution data of the different supports.
Response: For the characterization of supported metallic NP’s, not only the mean sizes of the catalyst as well as the distribution of the NP’s are important, which is why the presentation of distribution graphs are necessary. Indeed, as shown in Figure 3, the distribution of Au NP sizes can be quite large. Given that the Au NP’s are pre-formed in solution and independent of the support, we doubt that the pore size of the support will have much influence on the NP size, compared to the wetness-impregnation method, for instance. At this juncture, there appear to be a clear correlation of the particle size distribution with the available surface area (determined by BET analysis). The effect of pore size cannot be completely ruled out, but as we don’t have this data, this will be highly speculative at best.
2) In the case of Au/C it should report the pore size distribution data of the support C. This could be usefull for a best understanding of the relative lower catalytic activity than the metal oxyde supports. This is due to (probably) a significant amount of the adsorbed Au NP´s should be placed into very narrow pores which either difficults the acces of the substrated to them or where these suffer from severe sterical restriction
Response: We do not have the pore size distribution of any of the supports we used in this work. We attributed the lack of reactivity to the inability of the support to bind hydrogen (Fig
3) Concerning the data of Table 4, (on the basis of hypothetical nature of the interaction of Au with the different supports) some tentative explanation should be done about larger agglomeration of Au/C NP´s relative to metal oxyde supports.
Response: This part of the discussion had been supplemented.
4) Neither in section 2.2 nor in Experimental section it is clarified if 30 min. in the kinetic studies was the equilibrium times. Differences in % conversion of Table 2 should reflect, in each of the cases, that different factors affect the mechanisms of the studied reaction determining different equilibrium times; e.g. in case of Au/C, expected adsorptivity of the aromatic substrates on activated carbon marks an important difference with metal oxide supports. This point deserves a short discussion the author should include in the manuscript.
Response: 30 min is the total reaction time. Note that the reaction was performed in a batch reactor, so there is no ‘equilibrium time’ (I assume the reviewer refer to steady state conversions that can be obtained in continuous flow?).
Minor questions:
The following text of rows 20-21, "Secondary amines are mostly derived from primary amines, either by the reaction with a reactive alkyl halide or reductive amination with a carbonyl compound(Scheme 1, routes 1 and 2)" does not fully match with Scheme 1. Please revise.
Response: We thank the reviewer for spotting the mistake – Scheme 1 has been revised to include the alkyl halid.
In the text at foot of Table 1, the superscript 1 should be changed by a one, isn´t it?
Response: Indeed – this is a typo, which is corrected in the revised manuscript.
On discussing the adsorptivity of Au (soft Lëwis acid) on AC, arene centres existing in the surface of the last, perhaps should be considered to act as active adsorption sites.
Response: This is a good point. This has been incorporated into the revised manuscript.
TEM images used for the counting of the sizes distribution of Au NP´s should be provided as supplementary information
Response: We were unable to retrieve the images (recorded 8 years ago), when the first author was a PhD student at Imperial College. Some of the TEM images were recorded at the industrial collabroators site, and some are collected at Imperial College. Since then, the Chemistry department has moved from South Kensington to a new campus in White City, and unfortunately the record had been lost during the move.
Round 2
Reviewer 1 Report
Accept in present form.
Moderate editing of English language required.
Reviewer 2 Report
I recommend publication of the revised version of this manuscript